# The Role of CDK Pathway Dysregulation and Its Therapeutic Potential in Soft Tissue Sarcoma

**DOI:** 10.3390/cancers14143380

**Published:** 2022-07-12

**Authors:** Johannes Tobias Thiel, Adrien Daigeler, Jonas Kolbenschlag, Katarzyna Rachunek, Sebastian Hoffmann

**Affiliations:** Department of Hand, Plastic, Reconstructive and Burn Surgery, BG Unfallklinik Tuebingen, University of Tuebingen, 72076 Tuebingen, Germany; adaigeler@bgu-tuebingen.de (A.D.); jkolbenschlag@bgu-tuebingen.de (J.K.); krachunek@bgu-tuebingen.de (K.R.); shoffmann@bgu-tuebingen.de (S.H.)

**Keywords:** CDK, cyclin-dependent kinase, sarcoma, soft tissue sarcoma, CDK inhibitors, CKI

## Abstract

**Simple Summary:**

Soft tissue sarcomas (STSs) are rare malignant conditions with more than 70 subtypes that are difficult to treat, especially in advanced or metastatic states. Recently, next-generation sequencing technologies have provided comprehensive information and developed personalized medicine for treating cancer in general and STSs in particular. Growing knowledge of diverse gene alterations and biomolecular targets in various subtypes of STSs raises hope for novel treatment approaches and heralds a paradigm shift in the treatment of STSs. Activated cyclin-dependent kinases (CDKs) appear to play a critical role in sarcoma development and represent important targets for sarcoma therapy. This review discusses how CDK signaling influences STS development and its implications for STS prediction and targeted treatment.

**Abstract:**

Soft tissue sarcomas (STSs) are tumors that are challenging to treat due to their pathologic and molecular heterogeneity and their tumor biology that is not yet fully understood. Recent research indicates that dysregulation of cyclin-dependent kinase (CDK) signaling pathways can be a strong driver of sarcogenesis. CDKs are enzyme forms that play a crucial role in cell-cycle control and transcription. They belong to the protein kinases group and to the serine/threonine kinases subgroup. Recently identified CDK/cyclin complexes and established CDK/cyclin complexes that regulate the cell cycle are involved in the regulation of gene expression through phosphorylation of critical components of transcription and pre-mRNA processing mechanisms. The current and continually growing body of data shows that CDKs play a decisive role in tumor development and are involved in the proliferation and growth of sarcoma cells. Since the abnormal expression or activation of large numbers of CDKs is considered to be characteristic of cancer development and progression, dysregulation of the CDK signaling pathways occurs in many subtypes of STSs. This review discusses how reversal and regulation can be achieved with new therapeutics and summarizes the current evidence from studies regarding CDK modulation for STS treatment.

## 1. Introduction

Soft tissue sarcomas (STSs) are rare, heterogeneous malignant tumors that accounts for about 1–2% of all cancers. In the United States, there are 12,750 new cases diagnosed yearly, and STSs kill 5270 people each year [1]. The crude incidence rate of STSs is 4.71 per 100,000 people in Europe, with an estimated 25,851 new cases in the European Union [2]. Soft tissue sarcoma is currently composed of approximately 80 subtypes defined by the World Health Organization (WHO), classified based on a combination of unique morphological, immunohistochemical, and molecular characteristics [3]. Although the ultimate cellular origin of sarcoma subtypes remains unclear, there is increasing evidence that they arise de novo from mesenchymal pluripotent stem cells [4,5].

The mainstay of therapy has been surgical resection with negative margins, but the prognostic impact of tumor-free margins on prognosis remains controversial [6,7]. The risk of recurrence and distant metastasis (DM) is mainly related to tumor biology. There are significant variations in the incidence of DM across different sarcoma histologies, and the tumor grade and size impact this risk significantly in STSs [8]. Overall, the estimated five-year survival for STSs is ~57–62% and can vary widely depending on the disease stage and the complex interplay between the anatomical site and STS subtype [9]. Patients with advanced STSs have a median overall survival of fewer than 18 months and require systemic therapies, which unfortunately have not been very promising so far [10,11,12].

Additionally, due to the fact that STSs are heterogeneous, responses to generalized therapy and active substances are variable and usually no longer translate among unique subtypes [13]. Therefore, the treatment for each sarcoma subtype should be individual and personalized. To accomplish this goal, biomarkers and critical points in the signaling pathways for growth and progression must be elucidated and characterized. Recent advances advise that changes in cyclin-dependent kinase (CDK) pathways are vital drivers of sarcomagenesis specifically and of cancer in general [9,10,14,15].

## 2. What Are Cyclin-Dependent Kinases (CDKs)?

Cyclin-dependent kinases (CDKs) were first discovered through genetic and biochemical studies in various organisms, including yeasts and complex organisms such as frogs. With their discovery came a growing understanding of the importance of CDKs in cell reproduction [16,17]. In the 1960s, the cell-cycle phase in eukaryotic cells was described as a sequence of four phases (see Figure 1). A few years later, in 1987, the first CDK was described, cell division cycle 2 (cdc2), again changing our understanding of cell-cycle progression. As scientists discovered cdc2 first, it was named CDK1 [18]. CDKs are serine-threonine kinases; they phosphorylate their substrates at serines and threonines. The enzymes regulate transcription and mRNA processing and may also be involved in neuronal differentiation [10]. CDKs have no function in resting cells because of a structural confirmation that obscures the catalytic and substrate-binding domains [10,19,20]. Their serine/threonine-specific catalytic core partners, called cyclins, inherit regulatory subunits, controlling kinase activity and substrate specificity [19]. Specific subsets of cyclins and CDKs regulate each phase transition in the cell cycle. Therefore, CDKs are essential enzymes that control the transition of the individual phases in the cell cycle through restriction points in a compassionate manner [21].

CDK–cyclin complex activity is tightly regulated by many CDK inhibitors (CKIs), which stop the cell-cycle progression under unfavorable conditions [22]. To date, 20 different CDKs (numbered from CDK1 to CDK20) and 29 human cyclins and cyclin-like proteins have been identified [18]. UniProtKB IDs list the functions, structures, sequences, and interactions of many known CDKs, accessible at the website https://www.uniprot.org/uniprot/ (accessed on 16 May 2022) [20].

CDKs were traditionally divided into two groups: CDKs of the first group can bind multiple cyclins and regulate the cell cycle progression (CDKs 1–4, 6 and 7). CDKs of the second group form complexes with a single cyclin and are involved in regulating transcription processes (CDKs 7–9,12, 13 and 19) [23,24,25,26]. CDKs 5, 10, 11, 14–18, and 20 do not fit into the abovementioned categories. They lack explicit functional annotations and have different functionalities, which are often tissue specific [23]. CDK5, for example, cannot directly control cell-cycle regulation [27]. It regulates neuronal development and post-mitotic neuronal activities by binding with p35 [28]. Substrates of CDK5, such as transcription factor p53 and myocyte enhancer factor 2 (MEF2), are involved in sarcoma progression [29,30].

Due to deregulation of the CDK pathway, uncontrolled cell proliferation often leads to cancer [31].

There is increasing evidence that the impaired activation and expression of CDKs are associated with tumors; conversely, targeting CDKs in tumor cells has become a promising therapeutic strategy [32]. The inhibition of CDKs can reduce the growth and progression of sarcoma cells and lead the diseased cells into apoptosis [20,32].

## 3. Selected CDKs and Their Role in Sarcoma Research and Treatment

CDKs are focused on providing targeted therapy to patients suffering from sarcoma. However, the same treatment differs in efficacy in different patients and tumors. These results reflect the unique microenvironment of each tumor. In addition to the specific microenvironment, compensatory pathways that undermine the mechanism of new CDK-related treatments have also been discovered [9,33]. Based on sequence homology, scientists have mapped and grouped CDKs, cyclins, and CKIs. As more and more information has been collected, it has become clear that the earlier, rather strict criteria for classifying these proteins are no longer correct. Recently, studies have shown that complexes of CDK and cyclin subunits are themselves highly active [19]. In the following section, we summarize the most important CDKs in STSs, potential biomarkers, and possible molecular targets of CKIs.

### 3.1. CDK1

CDK1 (CDC2) plays a vital role during the cell cycle. This enzyme strongly regulates the S phase and the G2 phase. The separate binding of cyclin A and cyclin B to CDK1 drives the transition from the G2 phase to the M phase [32,34]. Experiments with knockout CDK1 mice have shown that CDK1 is essential for initiating mitosis [34]. The phosphorylation of the complex of CDK1 and cyclin B by Wee1, a serine/threonine kinase, leads to the inhibition of CDK1 (see Figure 1) [32]. Therefore, by inhibiting the inhibitor Wee1, the activity of CDK1 can be increased. The proof of this principle has been shown using the CKI MK1775 (*adavosertib*), a Wee1 inhibitor in different cell lines, derived from human liposarcomas (LPSs) and from rhabdomyosarcomas (RMSs). Notably, in these cell lines, CDK1 is strongly expressed during the progression of the S phase and the transition from the G phase to the M phase. The proliferation ability of the cells is decreased by the inhibition of CDK1 expression or activation [35,36].

### 3.2. CDK2

CDK2, similar to CDK1, is a serine/threonine kinase and is involved in the transition from the G1 phase to the S phase and is closely associated with cyclins A and E (see Figure 1). For the treatment of STSs, p27, a tumor suppressor protein, is of interest. P27 inhibits CDK2. Therefore, upregulation of p27 in a human RMS cell line results in potent inhibition of CDK2, and decreases the proliferation ability of cells [37,38]. Transforming growth factor-beta (TGFβ1) was found to initiate the upregulation of p27 in RMS. It also enhances the binding affinity of p27 to the complex of CDK2 and cyclin E [37]. Due to these two mechanisms, TGFβ1 is a promising inhibitor of tumor cell proliferation in RMS, round liposarcomas, and myxoid cell lines. The cell line HS-18 derived from human LPSs highly expresses CDK2 and cyclin A and cyclin E [39,40]. This high expression of CDK2 and genetic aberrations in the coding sequences for CDK2 in sarcoma have also been associated with a bad clinical course. Therefore, CDK2 gene aberrations are considered crucial prognoses influencing factors [41].

### 3.3. CDK4 and CDK6

CDK4 and CDK6 both interact with D-type cyclins. Three D-type cyclins are currently known: cyclins D1, D2, and D3. Not only are binding partners identical, but also 71% of the amino acid identity in CDK4 and CDK6 is the same [32]. The crucial role of both proteins is to promote the progression of the G1 phase and the transition from the G1 phase to the S phase (see Figure 1).

P16 (also known as CDK inhibitor 2A) and p21 (also known as CDK inhibitor 1A) are familiar CKIs for CDK4 and CDK6; both p proteins are tumor suppressors. CDK4 and CDK6 can phosphorylate the retinoblastoma tumor suppressor protein (Rb1). In this way, both CDKs inactivate Rb1 and silence multiple genes [42].

The interplay of Rb1, CDK4, and CDK6 influences cancer cell proliferation, differentiation, and transformation [43]. In sarcomas, research on CDK4 and CDK6 as well as CKIs such as *palbociclib* is still ongoing and the most promising, even if the majority of studies are experimental. *Palbociclib* is a potent inhibitor (CKI) of both CDK4 and CDK6. Rb1-proficient ovarian cancer cell lines are sensitive to *palbociclib*; in contrast, glioblastoma multiforme cell lines are highly resistant to *palbociclib* [44,45]. These cell lines inherit deletions or mutations in the Rb1 gene. This highlights that active, hypophosphorylated Rb1 is key to the efficacy of *palbociclib*. By confronting Rb1-deficient cell lines with extremely high levels of CKI, the principle was proven. Even high concentrations of *palbociclib* have failed to induce G1 arrest [45]. Some authors have suggested that Rb1 is a predictive biomarker for response to CDK4 and CDK6 inhibitors [32].

A total of 85% of myxoid and round cell liposarcoma highly express both CDKs. Rb1 immunoreactivity has been reported in 66% of these sarcoma subtypes [46]. Scientists have observed a significant overexpression of CDK4 and CDK6 in mouse model, linked with the progression and occurrence of both dedifferentiated (DDLPSs) and well-differentiated liposarcomas (WDLPSs) [47,48].

CDK4 may even be used as a prognostic marker, as poor disease-specific survival was associated with high expression of CDK4 in 56 patients with LPSs [49]. Additionally, a significantly high expression of CDK4 has been found in patients with WDLPSs recurrence after surgery [49]. Complementary low expression of CDK4 in these tumors was associated with a better prognosis and a higher progression-free survival [50].

In addition to *palbociclib*, *ribociclib* is another interesting CKI, also known as LEE011, and a selective CDK4 inhibitor. This drug arrests liposarcoma tumor cells in the G0 phase and G1 phase after 24 h of incubation and limits tumor cell proliferation [51]. Few clinical trials have been designed with this drug, and the majority of registered clinical trials are still ongoing. Recently, a phase Ib study in patients with LPSs and *ribociclib* has been completed [52]. Together with other clinical trial results, these are discussed in Section 5. CDK4 and CDK6 are very similar from the ultrastructural point of view. Interestingly, the efficacy of dual CDK4/6 inhibitors against CDK4 and CDK6 is different. In vitro studies have found that the dual inhibitors *abemaciclib*, *ribociclib*, and *trilaciclib* were more powerfully inhibiting CDK4 than CDK6, while *palbociclib*, in contrast, had comparable efficacy against both CDK4 and CDK6 [53,54,55]. Finally, *abemaciclib* and *trilaciclib* not only inhibit CDK4/6 but also have a slight inhibitory effect on CDK5 and CDK9.

In RMS, CDK4/6 inhibitors appear to be able to arrest tumor growth. CDK4 knockdown mice receiving RMS cell lines showed impaired proliferation and poor transformation of tumor cells arrested in G1 phase. Deficient Rb1 phosphorylation induced this arrest [56]. In 2015, RMS expressing low levels of CDK4 were shown to be especially sensitive to ribociclib and, therefore, the inhibition of CDK4/6 [57].

In summary, there are few trials on CDK4 and CDK6 inhibitors in the treatment of STS, most of which are in the preclinical or early clinical stage [32,58,59,60,61].

### 3.4. CDK9

CDK9 is the catalytic subunit of two enzymes: one is Tat-activating kinase and the other is positive transcription elongation factor b (pTEFb) [61]. CDK9 is present and expressed during the whole cell cycle [62]. CDK9 supplies the transcriptional homeostasis and, therefore, fundamental regulation of gene transcription [63]. In malignancies, physiological homeostasis of transcription is generally absent; oncogenes take control and modulate transcription. Therefore, CDK9 is of great interest for targeted-therapy concepts because of its function as a “guardian of cellular transcriptional homeostasis” [20,32,64,65]. The interaction of CDK1/CDK2 and CDK9 can lead to apoptotic arrest in G2 and M phases causing inhibition of the whole cell cycle [66].

Alterations in CDK9 activity in RMS are suspected to inhibit the physiological differentiation of tumor cells [65,67,68]. Synovial sarcoma also shows a clear correlation between poor prognosis and high levels of CDK9 [69]. In chronic lymphocytic leukemia, small-cell lung cancer, and breast cancer, two CDK9 inhibitors (*alvocidib* (*flavopiridol*) and *seliciclib*) have already been applied [70,71].

### 3.5. CDK11

CDK11 differs from other CDKs; it is not only encoded by a single gene but by two genes. CDC2L1 (CDK11B) and CDC2L2 (CDK11A) share many bases homologously. CDC2L2 is specific to humans and is absent in mice [72,73]; both conventional groups of CDKs do not apply to CDK11 due to its variety of tasks [74]. In eukaryotic cells, a minimum of ten isoforms have been cloned already. The most common and highly active of them is CDK11^p110^ [75,76,77]. CDK11^p110^, similar to all CDK11 isoforms, is associated with RNA splicing, transcriptional regulation, and cell division [78]. Although very similar, the isoforms slightly vary in their way of functioning. CDK11^p110^ forms a complex with cyclin L and mainly interacts with RNA processing and transcription, whereas CDK11^p58^ pushes mitosis and acts kinase specific in the G2 and M phase [75,78,79,80,81].

LPS tissue microarrays analyzed by immunohistochemistry showed high levels of CDK11. Benign lipoma tissue, in contrast, expressed significantly less CDK11 [74]. First attempts with synthetic lentiviral shRNA and siRNA suppressing CDKs successfully induced and increased doxorubicin’s cytotoxic capability in LPS cells [74]. However, CDK 11 isotypes’ functions cannot be simplified. Scientists have proven the antagonistic effects of CDK11^p58^ and CDK11^p110^ in breast and prostate cancer. CDK11^p58^ appears to induce anti-metastatic and anti-proliferation effects. In contrast, CDK11^p110^ promotes the cell viability and survival abilities of cell clones [82,83,84].

Although research has intensified on CDK11 inhibitors, no single CKI has been designed yet.

## 4. CDK Pathway Dysregulation in Prevalent STSs

The following section provides further insight into the dysregulation of the CDK pathways, characterizes potential future molecular targets, and highlights the preclinical studies of the three most common (extra abdominal) STS subtypes in adults. In addition, two examples for rarer subtypes such as malignant peripheral nerve sheath tumors (MPNSTs) and myxofibrosarcomas (MFSs) are addressed and summarized.

### 4.1. Liposarcoma

Liposarcomas (LPSs) arise from adipocytes and account for a significant proportion (~13–20%) of adult sarcomas [10]. Three biological groups depend on the molecular profile and growth characteristics. The groups consist of well-differentiated (WDLPSs) and dedifferentiated liposarcomas (DDLPSs), myxoid/round cell LPS (MCRLPSs), and pleiomorphic LPSs (PLPSs) [85,86]. However, WDLPSs/DDLPSs account for around 60% of all LPSs, while PLPSs are the rarest (~5%) [87]. WDLPSs/DDLPSs are two sides of one subtype. Unlike WDLPSs, DDLPSs can multiply quickly, are aggressive, and metastasize early [88]. They both involve the amplification and overexpression of CDK4, HMGA2 (high-mobility group AT-hook 2), and mouse double minute 2 (MDM2) due to an amplification of chromosome 12q13-15. In contrast, over 95% of MCRLPSs carry a translocation of FUS and DDIT3 (CHOP) genes, whereas PLPSs often causing the loss of tumor suppressors p53 and Rb1 [89,90,91].

Treatment of locally advanced WDLPSs and DDLPSs or systemic disease is complex because these tumors are hardly chemosensitive [92,93,94]. Current clinical trials are focusing on MDM2 inhibitors, CDK4, and CDK6. Some of these trials are discussed later in Section 5. A total of 90–95% of all WDLPSs/DDLPSs have high levels of CDK4 and almost 100% have co-amplification of MDM2 and, in theory, represent a very good target for new drugs [52,86]. *Palbociclib* efficiently inhibits CDK4 and CDK6, and thus, cell growth of WDLPSs/DDLPSs in vitro and in xenograft models [91]. Feeding the CDK4/6 inhibitor *ribociclib* to mice bearing human LPS xenografts ultimately decreased tumor biomarkers, including Rb1 phosphorylation. Continued treatment inhibited the growth of tumors or even caused a regression [51].

CDK11 (see Section 3.5) is suggested to generally be crucial for the growth and proliferation of LPS cells. While CDK11 expression is significantly higher in LPSs, no CDK11 changes are detected in benign lesions (lipomas) [74]. CDK11 could be a promising therapeutic target for the treatment of LPSs. However, there have been no clinical trials of targeted CDK11 inhibition in LPSs due to the lack of a specific CKI.

### 4.2. Leiomyosarcoma

Leiomyosarcoma (LMS) is a neoplasm characterized by smooth-muscle differentiation, the loss of tumor suppressors, and a lack of recurrent driver mutations. It accounts for up to 20% of all sarcoma diagnoses [95,96]. As typical for most STSs, the overall frequency of LMS increases with age, peaking at 70. However, LMS sometimes also occurs in younger patients starting from 30 years on. Unfortunately, the lack of correspondence of established LMS cell lines to the original mesenchymal neoplasm limits understanding of this subtype.

Nevertheless, some preclinical models have already been developed [95]. The loss of tumor suppressor gene Rb1 leads to a lack of cell-cycle regulation at the checkpoint from the G1 phase to the S phase, leading to uncontrolled cell division [97]. Up to 90% of LMS patients hold altered Rb-cyclin D1 signaling pathways, resulting in a fatal prognosis, which could be an essential factor for proliferation, at least in a subgroup of LMS [98,99]. In three different LMS cell lines, seliciclib, a pan inhibitor of CDK1, 2, 5, and 9 combined with cisplatin and single use, caused a decline in CDK2 mRNA and protein concentration over 72 h [100,101].

Deletions and mutations of Tp53 occur in roughly 50% of cases [10]. Riva et al. presented a case of ten different LMSs affecting a single patient over three years [102]. All 10 LMSs had a deletion on chromosome p19 that inherits the gene for cyclin-dependent kinase inhibitor 2A (CDKN2A). Downregulation of CDKN2A has been described in many malignant processes and significantly correlates with shorter patient survival [102,103,104].

### 4.3. Undifferentiated Pleomorphic Sarcomas

Undifferentiated pleomorphic sarcomas (UPSs) are a common STS histotype. They account for 14% of all adult STSs, and about 60% occur in the extremities. Nevertheless, they can grow anywhere in the human body [105,106]. Up to 78% of UPS tumors have a deletion of the Rb1 gene due to a loss of regions within chr13q [107,108]. Furthermore, genetic mutation, deletion, or silencing of chr9p21, the region containing the CDK inhibitor p16, leads to free activation of cyclin D-CDK4/6 kinases. A total of 29% of UPSs upregulate genes encoding MDM2 and CDK4 [10,109]. The oncoprotein MDM2 ubiquitinates the tumor suppressor p53 and promotes its proteasomal degradation. Overexpression of MDM2 leads to downregulation of the CKI p21. P21 is a transcriptional target of p53, and its downregulation causes hyperactivation of CDKs [10]. The loss of p53 leads to general genetic instability and promotes additional tumor-promoting mutations. P53 plays a crucial role in the regulation of DNA repair, the cell cycle, programmed cell death, and cellular senescence [110].

Recently, it has been shown that UPSs often have copy-number alterations or mutations in the tumor suppressor genes Rb1 and Tp53. Deletion and mutation of the p53 gene, Tp53, can still be observed in 43% of all UPSs [111]. A recently published study revealed that Rb1- and p53-deficient UPSs required S phase kinase-associated protein 2 (Skp2) for survival. Skp2 can drive the proliferation of UPS cell lines by degrading p21 and p27. The loss of both Rb1 and p53 in patient-derived cell lines renders undifferentiated pleomorphic sarcoma dependent on Skp2, which could provide the basis for promising novel systemic therapies [112].

Further studies have verified that targeting the neurofibromin 1 (NF1) gene and a deletion of CDKN2A can lead to the formation of UPSs [113,114,115]. Despite the high number of experimental studies published so far, there is not a single clinical trial yet that has investigated potential CKIs in patients suffering from UPSs.

### 4.4. Myxofibrosarcomas

Myxofibrosarcomas (MFSs) are rare mesenchymal soft tissue sarcomas with a high local recurrence (LR) rate [116]. About 5% of all sarcomas are MFSs, the myxoid component in combination with a hypocellular appearance characterizes these sarcomas [117]. A study in 2022 revealed a 5-year LR of 12%, metastasis of 17%, and overall survival (OS) of 84% in 293 patients [118]. Whole exome sequencing of nearly 100 MFS tumors revealed frequent alterations in genes related to the tumor suppressors p53, p15, p16, and Rb1, in addition to MDM2, cyclin D1, and CDK6 [10,119]. These mutations led to the inactivation of the NF1 gene. UPSs showing this inactivation enhance Ras signaling and upregulation of cyclin D1 transcription [91,120].

Amplifying the CDK6 gene and overexpression of the protein leads to higher grading, highlighting the clinical importance of this molecular aberration in promoting disease progression in MFSs [121]. Li et al. characterized the relevance of the alpha-methylacyl coenzyme A racemase (AMACR) in MFSs. AMACR protein overexpression and gene amplification was associated with less metastasis-free survival and disease-specific survival [122]. Stable AMACR knockdown suppresses cell proliferation, growth, and cyclins D1 and T2 expression [122]. Furthermore, downregulation of CDK2 induces high tumor suppressor p12 levels, thereby, inducing MFS cell cycle arrest and apoptosis. Vice versa, a low p12 level is a poor prognostic factor in patients with MFSs [123]. So far, unfortunately, the aforementioned findings and knowledge about potential targets such as CDK6 or CDK2 have not resulted in a clinical trial in MFSs.

### 4.5. Malignant Peripheral Nerve Sheath Tumors

Malignant peripheral nerve sheath tumors (MPNSTs) are aggressive sarcomas that develop in the connective tissue surrounding the nerves and occur predominantly in the extremities [124]. MPNSTs account for 3–10% of all STS diagnoses and arise sporadically (in ~50% of cases) or in patients with the cancer predisposition syndrome, neurofibromatosis type I (NF1) [108,125]. MPNSTs constitute a significant cause of fatal outcomes in NF1 patients, with a 5-year survival rate of only 20–35%. In most MPNSTs, the loss of the tumor suppressor proteins p16 and p27 leads to overexpression of CDK2 and CDK4/6 [126]. These changes results in the inactivation of the Rb1 tumor suppressor, one of the most important tumor suppressors in human cancers, which is a meaningful event in the pathogenesis of MPNST [125,127,128,129].

Characteristic for MPNSTs are NF1-inactivating mutations and frequent genetic disorders of the tumor suppressor gene CDKN2A and polycomb repressor complex 2 (PRC2) [130,131]. In addition, several genes of the CDK metabolism, such as the genes for cyclins D1 (CCND1) and E1 (CCNE1), are suppressed by PRC2 [132,133]. Many genetically engineered mouse models have been developed to more fundamentally understand the genetic changes that occur during MPNST development. For example, mutations of NF1 and Trp53 or NF1 with CDKN2A are possible drivers of MPNST development [134,135,136].

Genomic analyses of MPNSTs have demonstrated the loss of the CDKN2A/B locus in up to 70–80% of MPNSTs [137,138,139]. The loss of CDKN2A leads to the upregulation of CDK4/6, which causes the initiation of the S phase and promotion of mitosis [125,140,141,142]. However, not all MPNSTs are CDKN2A inactivated. In some MPNSTs, only a heterozygous loss is present, which does not cause total inactivation [10,130,143]. Nevertheless, the loss of CDKN2A leads to a more potent activation of CDK4/6, suggesting CDK4/6 inhibitors (CKIs) to be potentially effective in the therapy for MPNSTs [138,144]. Again, clinical studies and evidence are lacking.

In 2020, Kohlmeyer et al. studied and reported a new oncoprotein and negative regulator of Rb1, p53 signaling, and its role in MPNST cell lines [126]. RAB-like 6 isoform A (RABL6A) is this novel protein [145,146,147,148]. Conversely, the silencing of RABL6A led to MPNST cell death and cell-cycle arrest in the G1 phase. Crucial elements for this arrest are the upregulation of p27, the downregulation of CDK4/6, and the activation of Rb1 [126]. In vitro *palbociclib* promoted MPNST cell death via the reactivation of Rb1 (in a RABL6A-dependent manner). In vivo, it suppressed MPNST. The antitumor effect was enhanced by low-dose combinations of drugs that inhibit multiple kinases (CDK4/6, CDK2). The combined therapy of multiple drugs targeting different signaling pathways could effectively treat MPNSTs [124,126].

### 4.6. Preclinical Findings in Other Rare Subtypes

BCL6 corepressor (BCOR) sarcomas are rare and defined by alterations of the BCOR gene. These alterations are caused by fusion or BCOR intern tandem duplication. In a retrospective database research, 40 uterine sarcomas in 1390 patients were identified to hold BCOR gene rearrangements [149]. Furthermore, 38% of these uterine BCOR sarcomas showed amplification of CDK4, whereas 45% enhanced MDM2. Finally, 28% were positive for the homozygous deletion of CDKN2A, and therefore, lacked the CDK4 inhibitor. In this regard, a rare clinical case of a BCOR-CCNB3 fusion sarcoma (BCS) treated with palbociclib is discussed in Section 5.

## 5. Lessons from Clinical Trials Regarding CDK-Directed Therapy in STSs

CDKs appear to be promising targets for cancer treatment and a range of non-oncological diseases such as autoimmune diseases, inflammatory diseases, viral infections including COVID-19, and central nervous system diseases [20]. In recent years, a steadily growing number of clinical trials with CKIs as single agents or in combination with other drugs underlines the importance of CKIs in sarcoma research. Although extensive efforts have been made to discover CKIs, only four CDK inhibitors have been approved since the early 1990s, all dual selective CDK4/6 inhibitors [150]. Yet, clinicians have apply *palbociclib*, *ribociclib*, *abemaciclib*, and *trilaciclib* for treating estrogen receptor-positive/HER2-negative metastatic breast cancer [151,152,153,154]. Most CKIs under preclinical and clinical investigation targets the cell cycle-associated subset traditionally grouped as CDK group 1. In contrast, the development of CKIs selectively targeting CDKs that regulate transcription (CDK group 2 such as CDK9 or non-groupable CDKs) has just begun.

Targeted therapy with high CDK selectivity has become a significant trend. In contrast to pan-CKIs, selective CKIs can avoid undesirable side effects. However, a significant disadvantage of selective inhibition is that, due to the very dense CDK network and multiple interactions, CDKs can compensate one another in their function and regulation of signaling pathways. For example, dual inhibition of CDK4 and CDK6 has little clinical impact on colorectal cancer and melanoma, as other CDKs such as CDK1–3 compensate for the loss of CDK4 and CDK6 [34]. These results indicate that some cancers might not be helpful to treat with a single specific CKI, which inhibits CDKs selectively.

Nevertheless, what do these preclinical results mean for STS therapy? Can these findings be transferred from the laboratory to the bedside? So far, no single CKI has been approved for therapy in STSs. Up to June 2022, more than a dozen clinical trials were registered with one or more interventional drugs for CDK-targeted therapy (see Table 1). The website www.clinicaltrials.gov (accessed on 30 June 2022) lists all updates in this field.

Most of the ongoing clinical trials are investigating the efficacy and side effects of CKIs. The focus lays mainly on CDK4/6 inhibitors in LPS, alone or combined with other agents. To the best of our knowledge, there are only two registered clinical trials (see Table 1) that have been completed and published yet [155,156,157].

### 5.1. Clinical Findings in LPSs

As previously stated, WDLPSs and DDLPSs are the subtypes of STSs that show the most promising results for treatment with CKIs, especially with CDK4/6 inhibitors [158].

In 2013, initial data of the first-ever reported phase II study on *palbociclib* showed a satisfactory progression-free rate in 30 patients suffering from WDLPSs/DDLPSs [156]. CDK4 amplification and Rb expression were closely monitored in this population of advanced tumor patients. All patients were treated with 200 mg *palbociclib* orally, once daily for 14 days in 21-day cycles. The treatment was generally well tolerated with no serious adverse events. Nevertheless, 24% of patients needed a dose reduction because of hematologic toxicity [156]. The 12-week progression-free survival (PFS) under *palbociclib* treatment was achieved at 66% with a median PFS of 17.9 weeks. PFS for the standard second-line treatment with *ifosfamide* was 65% and for *trabectin**,* 40–56% [159].

In a second non-randomized trial, aiming to reduce toxicity, an additional 30 patients were treated with 125 mg *palbociclib*, once daily for 21 days in a 28-day cycle. The most common side effect was reversible neutropenia. The median PFS was 18 weeks, similar to the previously described study. A 125 mg dose of *palbociclib* is commercially available and was FDA approved in 2015 for breast cancer [46,160,161,162]. In light of positive study results, *palbociclib* counts as category 2A evidence in the STS National Comprehensive Cancer Network (NCCN) guidelines; however, to date, it has not been approved by the FDA for STS indication [158].

An ongoing clinical trial (NCT02846987) with *abemaciclib* shows favorable PFS and objective tumor response (see Table 1). The already available preliminary results show toxicity to be manageable [163]. *Abemaciclib* belongs to the CDK4 and 6 inhibitor group and seems to be more promising in DDLPSs than *palbociclib*. In this study, 30 DDLPS patients received *abemaciclib* 200 mg continuously two times daily, and finally, 29 patients were evaluable for the primary endpoint; 76% of the patients survived after week 12, and the median PFS was 30.4 weeks. The response to *abemaciclib* seemed to be only partial. Major adverse events were anemia (37%), thrombocytopenia (17%), and diarrhea (7%).

These findings led to a recently initiated multicenter phase III trial (NCT04967521) testing *abemaciclib* in more patients with DDLPSs. The study focuses on response rate and PFSs in patients treated with *abemaciclib* for five years (see Table 1, last row).

*Ribociclib* is a more selective CDK4 inhibitor than *palbociclib* and also seems to be very promising. A proof-of-concept phase Ib trial published in March 2022 demonstrated that *ribociclib* paired with siremadlin, a p53-MDM2 antagonist, showed initial signs of antitumor activity in WDLPS or DDLPS patients [52]. In three groups, 74 patients received *siremadlin* and *ribociclib* in different schemes and dosages for 13 weeks (median, range 1–174): Group A (n = 26) went through a 4-week cycle of 15 mg *siremadlin* and 400 mg *ribociclib*, each once daily. Two weeks of treatment followed two weeks of pause. Group B (n = 29) involved the administration of *siremadlin* once every three weeks (range 120–200 mg) and *ribociclib* (range 200–400 mg) daily for two weeks, followed by one week of pause. Finally, in Group C (n = 19), the patients underwent treatment with *siremadlin* once every four weeks in different dosages (range 120–200 mg) and *ribociclib* (range 300–400 mg) every two weeks, followed by two weeks of pause. An important aim had been to reduce the risk of bone marrow toxicities. The 3-month PFS rates were 43.8% under regimen A, 65.9% under regimen B, and 55.6% under regimen C. The recommended dose of expansion (RDE) was determined as follows: *siremadlin* 120 mg every three weeks plus *ribociclib* 200 mg for two weeks followed by two weeks of pause (regimen B) [52]. In total, three patients achieved a partial response, 38 patients achieved stable disease, and one patient died due to hematotoxicity. Next-generation sequencing (NGS) showed p53 alterations in three of 74 patients, a negative predictor of response to MDM2 inhibitors [164]. These three patients lacked MDM2 amplification, two patients had a concomitant deletion of the Rb1 gene. This deletion is a robust negative predictor of CDK4 inhibitor response. Accordingly, to the gene aberrations, all three patients showed tumor progress despite treatment [165]. In summary, this recently published trial proved the low-dose daily regimen A to be less effective than the high-dose pulsed regimens B and C, at least treating advanced WDLPSs or DDLPSs in patients with MDM2 amplification.

In a previous study, the administration of MDM2 antagonists in combination with *doxorubicin* resulted in a high rate of hemotoxicity, precluding further development [166]. For this reason, the combination of an MDM2 antagonist with a targeted, less cytotoxic drug, such as selective CKIs such as *ribociclib*, may be a more relevant approach, and further studies culminating in a phase III study are needed [52].

### 5.2. Clinical Findings in LMS

Patients with uterus LMS (uLMS) harboring a CDKN2A mutation can profit from a treatment with *palbociclib*; 19% among 279 uLMS samples inherited mutations affecting the CDK pathway according to genomic analysis [167]. 

In one case, a woman with uLMS was treated with 125 mg *palbociclib* for 21 days monthly after multiple surgeries and frustrated chemotherapy. Before starting the CKI treatment, the patient faced metastasis and tumor progression. The tumor was proven to inherit a mutation of the CDKN2A gene, resulting in upregulation of CDK 4 and 6. While under *palbociclib* due to pancytopenia, the doses had to be reduced from 125 mg to 75 mg. After eight months of CKI treatment, the radiological follow-up by CT scan showed only minor enlargement of the LMS tumors; the tumor had not spread further [167].

A further retrospective NGS study on 114 patients with different sarcoma subtypes, found only 15 patients (13.2%) with relevant therapeutic targets validated by NGS [168]. Furthermore, only four of these 15 patients (26.7%) showed partial response or stable disease for more than six months. Although in one patient diagnosed with LMS holding a CDKN2A/B deletion, molecular profiling suggested a therapy with *palbociclib* and *fulvestrant*, no clinical efficacy was evident.

Based on these inconsistent findings, more extensive clinical trials evaluating CKIs, such as *palbociclib*, for treating LMS are highly required.

In 2017, clinicians in the United States initiated a two-armed study on advanced DDLPSs and LMSs (NCT03114527, see also Table 1). In this study, patients who had received at least one prior systemic medication are put under the combinatory regimen of *ribociclib* and *everolimus*. The dose for *ribociclib* is 300 mg for three weeks with the following week pause, while the dose for *everolimus* is 2.5 mg in a 28-day cycle. To determine the response evaluation criteria in solid tumors (RECIST), CT or MRI diagnostics are run at several different time points (week 8, 16, 24, and every 12 weeks following). Combining the mTOR inhibitor (*everolimus*) with a CDK4/6 inhibitor (*ribociclib*) is a promising approach to connecting novel targeted therapy with an already established antitumor drug. Positive results could lead to a new therapeutic tool for STSs. The first results will be available approximately in late 2022.

### 5.3. Clinical Findings in Other STSs

A phase II trial included patients who suffered from complex malign tumors that had shown no response to conventional therapy. The primary endpoint was to stabilize the disease for 16 or more weeks. Prior to treatment with *ribociclib*, patients’ tumor analyzations monitored cyclin D1/D3 amplification, CDK4/CDK6 amplification, CDK4/CDK6 mutations, and p16 mutations. Tumors with amplification of the cyclin D1-CDK4/CDK6 pathway showed a response to the treatment with *ribociclib*. In summary, only 3 (23.1%) of 13 included STS patients (of a total of 105 included patients with diverse tumor entities) had a partial response. Nevertheless, the primary endpoint was generally not met. The authors of the 2019 study recommended further investigation and additional complementary therapies to *ribociclib* monotherapy [169].

In 2006, a phase II study concluded that *flavopiridol* (*alvocidib*) had few, manageable side effects in treating STSs. Nevertheless, the pan-CDK2, CDK4, CDK6, and CDK9 inhibitor showed no objective treatment response [170]. Therefore, the authors did not recommend further studies with *flavopiridol* as monotherapy. However, earlier studies have shown additive benefits of (pan-)CKIs such as *flavopiridol*, improving the efficiencies of other drugs [10]. For example, the combination of *flavopiridol* and *doxorubicin* has been well-tolerated in vitro, and in phase I studies on LPSs and MPNSTs [100,101,171].

Two registered clinical trials (see Table 1) with novel selective CDK9 inhibitors (TP-1287 and PRT2527) are currently ongoing.

Finally, *palbociclib* administration resulted in a complete response in a rare case of refractory pediatric BCS [172]. Several genes in the CDK4/6-RB pathway had been overexpressed, making *palbociclib* an optimal therapeutic candidate for which child-specific dosing information was available. The targeted therapy with *palbociclib*, based on a dedicated germline and somatic whole-genome DNA sequencing combined with RNA sequencing, started at the age of eight. For 25 months, there was no further tumor evidence on imaging morphology.

## 6. Conclusions

Growing evidence on signaling pathways, microenvironments, and interactions among the increasing number of CDKs, CKIs, and tumor cells underlines that cyclin-dependent kinases are vital in sarcoma biology. Strikingly, the plethora of predominantly experimental studies is opaque. Suitable results to date are limited, and the abundance of (experimental) data can be confusing. However, clinical research is still in its infancy, heading for the ambitious goal of targeted-tumor therapy based on each patient’s biomolecular CDK-related footprint. The main focus in CDK clinical research, after extensively characterizing CDKs and their pathways, is now to create and test suitable CKIs and evaluate their efficacy as well as their side effects in clinical trials. For example, CDK11 could be a potential target in LPS therapy; however, so far, a CDK11 inhibitor is not available. Due to their tumor-inhibiting effect in multiple preclinical studies, CDK4/6 inhibitors have become the central component in phase 1 and 2 trials for various sarcoma subtypes. CDK4/6 inhibitors such as *palbociclib*, *abemaciclib* and *ribociclib* currently appear to have the greatest potential for future individualized sarcoma therapy and approval. CKIs have shown mixed results across STS subtypes, suggesting that new, more reliable biomarkers for sensitivity and resistance should be identified. Recent data show that microRNA related to CDK4/6 inhibitors may be useful as predictive biomarkers without compromising sensitivity to the treatment itself [173]. Nevertheless, scientists and clinicians have initially used CKIs only as single agents. However, the current trend is to use CKIs in combination with other chemotherapeutic agents or inhibitors. One advantage of these different combinations, which interfere with different signaling pathways of the STS cell, seems to be the minimization of drug resistance. Ultimately, further phase 2 studies and especially phase 3 trials are needed to confirm the clinical efficacy of CDK inhibitors alone or in combination in the treatment of STSs.

## Figures and Tables

**Figure 1 cancers-14-03380-f001:**
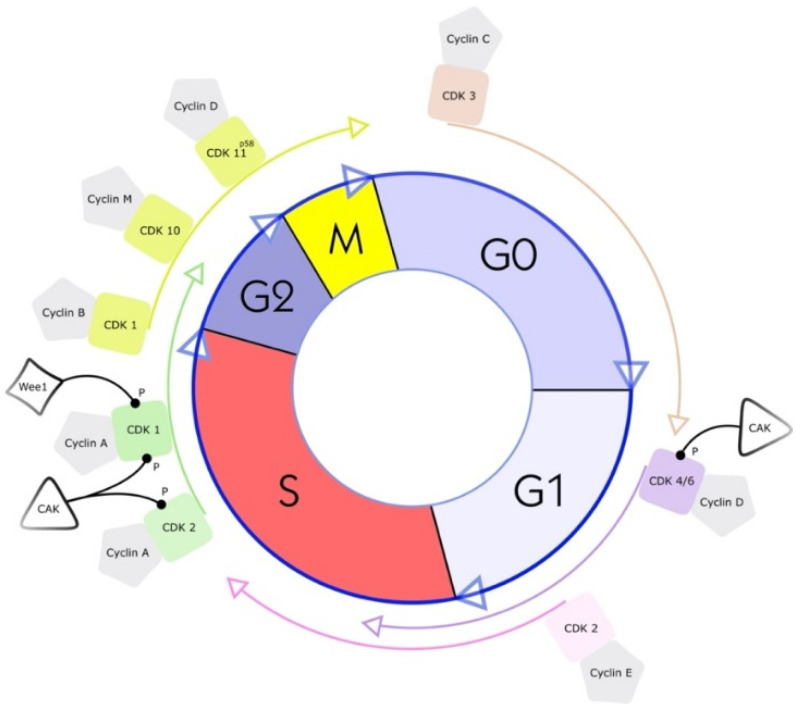
Cell cycle and a simplified illustration of interactions with selected kinases and cyclins. G1 is the cell cycle phase; in this phase, the cell increases in volume, but mitosis has not yet taken place. It is the first part of the interphase and transition into the S phase, the cell cycle’s replication phase. DNA replication takes place in this phase. The S phase (2nd phase) is usually constant in time (about 7 to 8 h) and lies between the G1 and G2 phases. Third, the G2 phase of the cell cycle is the second part of the interphase. It follows the S phase and enters the prophase of mitosis, in which the cell’s chromatin condenses into chromosomes. The following M phase describes the division of the cell. The nucleus splits (mitosis), and the entire cell divides (cytokinesis). Finally, the facultative G0 phase is the cell-cycle stage in which dormant or differentiated cells (e.g., nerve and muscle cells) are found [18].

**Table 1 cancers-14-03380-t001:** Registered ongoing and completed clinical trials with CDK-targeted therapy in STSs (www.clinicaltrials.gov (accessed on 30 June 2022)).

Status	Study Type	STS Type	CDK Target	Drug(s)	Estimated Enrollments	Identifier
Recruiting	Single-arm, single-institution, open-label, prospective phase II trial	LPS	CDK4/6	Ribociclib	30 participants	NCT03096912
Unknown	Single-arm, single-institution, open-label, prospective phase II trial	All STSs, LPS excluded	CDK4/6	Ribociclib	45 participants	NCT04040205
Recruiting	Phase I/II study	Kaposi sarcoma	CDK4/6	Abemaciclib	43 participants	NCT04941274
Unknown	Single-arm, single-institution, open-label, prospective phase II trial	LPS	CDK4/6	Ribociclib	30 participants	NCT02571829
Recruiting	Phase III study	All STSs and others	CDK4/6 with multiple others	Nilotinib, ceritinib, capmatinib, Palbociclib (and 8 more.)	960 participants	NCT03784014
Recruiting	Phase II study	LPS	CDK4/6 + anti-PD1	Palbociclib, INCMGA00012	42 participants	NCT04438824
Recruiting	Multicenter, open-label, dose-escalation phase I trial	Multiple sarcoma subtypes	CDK9	PRT2527	30 participants	NCT05159518
Recruiting	Phase I, open-label, multicenter, nonrandomized, multiple-dose, safety, tolerability, pharmaco-kinetic, and pharmaco-dynamic study	LPS	CDK4 + chemotherapy	PF-07220060, letrozole, fulvestrant	118 participants	NCT04557449
**Completed**	Phase Ib/II, open-label, multicenter study	LPS	CDK4/6 + MDM2	Siremadlin, ribociclib	74 participants	NCT02343172
Recruiting	Non-randomized, phase I/II study	All STSs and others	CDK2 + chemotherapy	BLU-222, carboplatin, ribociclib, fulvestrant	366 participants	NCT05252416
Recruiting	Phase Ib dose-escalation study	All STSs and others	CDK4/6 + chemotherapy	Abemaciclib, irinotecan, temozo-lomide	60 participants	NCT04238819
Recruiting	Non-randomized, open, two-cohort, phase II, multicenter national clinical trial. Twenty sites in Spain.	STSs and others	CDK4/6	Palbociclib	40 participants	NCT03242382
Recruiting	Phase I, open-label, dose-escalation, safety, pharmacokinetic, and pharmacodynamic study	STSs	CDK9	TP-1287	70 participants	NCT03604783
Active, not recruitung	Two-center, two-arm, phase II study	LPS	CDK4/6	Ribociclib, Everolimus	50 participants	NCT03114527
**Completed**	Phase II study	LPS	CDK4/6	Palbociclib	90 participants	NCT01209598
Active, not recruitung	Phase II study	DDLPS	CDK4/6	Abemaciclib	33 participants	NCT02846987
Recruiting	Phase III, multicenter, randomized double-blind study	DDLPS	CDK4/6	Abemaciclib vs. placebo	108 participants	NCT04967521

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
