# Peer review of "The Role of CDK Pathway Dysregulation and Its Therapeutic Potential in Soft Tissue Sarcoma"

_cancers, 2022, doi:10.3390/cancers14143380_

Round 1

Reviewer 1 Report

The manuscript herein presented is a an extensive review on the role of CDK and CDK - signaling mechanisms, role in normal cell proliferation, and how they can influence soft tissue sarcoma development;

This is a well structured review and with an exhaustive background, however, I believe that would be advisable to review Section 1 and Section 2 into a shorter version. Nonetheless, and in my view, this revision is not fundamental to accept this manuscript into publication, but, a shorter version on these topics would proportionate a more easy reading, and to highlight targeting CDK as a therapeutic option for soft tissue sarcomas.

Also, the authors should revise some confusing sentences along the text (as an exemple: see lines 68 and 69). 

Please, also revise formatation in order to uniformize letter font and size along the text. Furthermore we can find 3 sections numbered as Nº3 - Please revise.

Author Response

Dear editors, Dear Prof. J. Beier,
Thank you again for inviting us to this special edition "Soft and Hard Tissue Sarcoma
Current and Future Concepts in Diagnosis and Treatment".
We have revised the article as follows in response to the reviewer’s suggestions:
In particular, chapters one and two have been significantly shortened.
The abundance of experimental data, especially in section 2, has been shortened or
summarized.
Table 1 has been deleted. We have significantly shortened the general information in section
3, which is well known to the experienced reader, and focused more on clinical data for the
respective sarcomas.
The most important phase 1 and 2 studies to date are mentioned and discussed in more
detail. The former Table 2 (now Table 1) has been updated in terms of content, as new
studies have been listed and another one has been completed in the meantime. Sections 4
and 5 have also been revised with a focus on the clinical data and the opportunities and
possibilities for the future regarding CKIs in sarcomas.
Finally, an English-language correction was made again. This time at MDPI itself

Reviewer 2 Report

The manuscript from Thiel et al provides an extensive review about the role and mechanisms in which the CDK pathway participates in sarcomogenesis. The choice of the topic is an interesting and emerging topic for sarcomogenesis and as potential future targeted treatment approaches in sarcoma. A wide broad number of the literature is reviewed. Nevertheless, the review is in many parts too detailed, too general and loses the focus what is import for sarcoma. In conclusion, I cannot recommend publication in the present form.

Major concerns

- The plethora of data complicates the coherence and understanding. I clear red line with focus what is or might be important in sarcoma is clearly missing.

- Many parts describe extensively general information on CDKs, especially in the introduction as well as in Part 1 and 2 -or describe quite general information on sarcoma subtypes in part 3.

- So far, most findings on CDKs in sarcoma are based on pre-clinical data and positive results of clinical studies still limited. This should be critically discussed and how these results could be integrated in the clinical practice. It would be better to focus on the most advanced and thus convincing data, e.g. those from clinical studies. The experimental data are too thorough, without coherence and not concluding, as far as the pathway and the eventual role and applicability of these experimental therapies are concerned. In this way the manuscript fails to review the existing literature.

- Most clinical studies are only listed, but the results are not discussed.

- What is the information of table 1 for the role in STS?

- Again, the summary and conclusion are quite general and gives no outlook where might be the highest potential for CDK inhibitors in treatment of sarcoma, what are biomarkers or combination therapies.

Author Response

Dear Sir or Madam,

Thank you very much for your very detailed and constructive criticism.

We have revised the article as follows in response to your suggestion:

In particular, chapters one and two have been significantly shortened.

The abundance of experimental data, especially in section 2, has been shortened or summarized.

Table 1 has been deleted, you are correct. Too much information, lacking focus and a clear red line.

We have significantly shortened the general information in section 3, which is well known to the experienced reader, and focused more on clinical data for the respective sarcomas.

The most important phase 1 and 2 studies to date are mentioned and discussed in more detail. The former Table 2 (now Table 1) has been updated in terms of content, as new studies have been listed and another one has been completed in the meantime. Sections 4 and 5 have also been revised with a focus on the clinical data and the opportunities and possibilities for the future regarding CKIs in sarcomas.

Again, thank you very much for your extensive review!

Reviewer 3 Report

Thank you for choosing the topic "The Role of CDK Pathway Dysregulation and Its Therapeutic  Potential in Soft Tissue Sarcoma".

I think it's an interest topic.

My advice is to extensive edit the style, make the text shorter, highlighting the most relevant point.

A new revision after  this edititing is recommended

Author Response

Dear Sir or Madam,

Thank you very much for your stimulating review.

We have changed the following at your suggestion:

The text, especially the general information in chapters 1 and 2, has been significantly shortened and summarized

More focus has been placed on the current clinical data situation and, in our opinion, the manuscript has been comprehensively revised.

In addition, an English-language correction was made again. This time at MDPI itself.

Thank you again for your constructive criticism.

Round 2

Reviewer 2 Report

The manuscript from Thiel et al. aims to review the role and mechanisms in which the CDK pathway participates in pathogenesis of sarcoma. After the first revision, it is noticeable that many remarks and comments were taken into account and corrected, making the manuscript more comprehensive, coherent and target-oriented.

The introduction and the first part of the manuscript regarding the role of CDKs are shortened, as recommended, because the information was too extensive, general and irrelevant to the subject. 
The main part (section 3,4) also includes now more Phase I and II clinical trials and clinical data instead of preclinical, which makes the manuscript more convincing and achieves to shed focus on the role and applicability of CDK Inhibitors in sarcoma therapy.

Therefore, it has to be acknowledged the efforts the authors made to improve the review. However, some major and minor concerns remain:

Major concerns

- The manuscript would benefit further in my opinion, by more clearly distinguishing the pathway dysregulation and preclinical data from the clinical trials and data, because sections 3 and 4 remain not clearly demarcated with mixed discussion of preclinical and clinical data in both sections.

- In some parts, still the focus on the role of CDK pathway in sarcoma gets a bit lost, e.g. part 4 line 411-439. Here, further shortening should be considered.

- The authors should describe how they have selected the sarcoma subtypes on which they focus in their manuscript.

- In the section on liposarcoma, the data about abemaciclib in DDLPS of the phase II study should be discussed, which are more promising compared to palbociclib. At this, also the recruiting phase III study SARC041 should be discussed and mentioned. Both important studies also have to be mentioned in table 1.

- Regarding LMS, mentioning the one case report of LMS showing efficacy of palbociclib, they should also mention the case that was treated with a combination of fulvestrant and palbociclib without evidence of clinical efficacy.

- For DDLPS and LMS the SAR-096 trial should be mentioned and discussed.

- It might be worth to shortly discuss finding for rare subtypes, BCOR-CCNB3 fusion-positive sarcoma (doi: 10.1200/PO.19.00258; DOI: 10.1016/j.ygyno.2020.02.024)

Minor Concerns

- The paragraph on p7 l281-285 on a new cell line for MRLPS seems to be not relevant in this context and should be considered to be deleted with regard to focus the review on the topic of CDK pathway in sarcoma.

- The paragraph on p7 l291-292 should be changed to “WDLPS and DDLPS are the subtype of sarcoma that show most promising results for treatment with CDK4/6 inhibitors.”

- Regarding LMS, the study from Hemming et al (doi: 10.1158/1078-0432.CCR-21-3523) on preclinical model of LMS PDX and susceptibility to transcriptional CDK inhibitors might be discussed more detailed.

Author Response

Once again, many thanks for the extraordinarily structured feedback and constructive criticism. The following changes have been made:

  1. Preclinical results are now mentioned and discussed in section 3, while clinical results can now be found in detail in chapter 4.

  1. 411-439 has been significantly shortened.

  1. The introduction in chapter 3 has been clarified. We have limited ourselves to the 3 most common STS extra-abdominal and rarer ones such as MPNST or MFS in order to maintain structure and clarity.

  1. SARC041 and Abemaciclib in DDLPS have been added, both in the text and in the table.

  1. The case with fulvestrant and palbociclib has also been added to the document.

  1. SAR-096 was also added to chapter 4.

  1. BCOR sarcomas were added to chapter 3.6 and 4.

  1. Paragraph page 7 281-285 was deleted

  1. Paragraph page 7 291-292 was adjusted

  1. The study by Hemming et al. is discussed in more detail.

Once again, thank you very much for the outstanding comments and suggestions.

With warm regards, the authors.

Round 3

Reviewer 2 Report

The manuscript from Thiel et al. shows major improvements and it has clearly to be acknowledged the efforts the authors made to improve the review. I thank the authors for the constructive review process and their work for this manuscript on a rare disease. I recommend publication in Cancers.

This manuscript is a resubmission of an earlier submission. The following is a list of the peer review reports and author responses from that submission.